# Predicting Malnutrition Risk with Data from Routinely Measured Clinical Biochemical Diagnostic Tests in Free-Living Older Populations

**DOI:** 10.3390/nu13061883

**Published:** 2021-05-31

**Authors:** Saskia P. M. Truijen, Richard P. G. Hayhoe, Lee Hooper, Inez Schoenmakers, Alastair Forbes, Ailsa A. Welch

**Affiliations:** 1Department of Epidemiology and Public Health, Faculty of Medicine and Health Sciences, Norwich Medical School, University of East Anglia, Norwich NR4 7TJ, UK; s.truijen@uea.ac.uk (S.P.M.T.); richard.hayhoe@aru.ac.uk (R.P.G.H.); l.hooper@uea.ac.uk (L.H.); i.schoenmakers@uea.ac.uk (I.S.); alastair.forbes@uea.ac.uk (A.F.); 2School of Allied Health, Faculty of Health, Education, Medicine and Social Care, Anglia Ruskin University, Chelmsford CM1 1SQ, UK

**Keywords:** micronutrient deficiency biomarker, biochemical diagnostic tests, screening tool, undernutrition

## Abstract

Malnutrition (undernutrition) in older adults is often not diagnosed before its adverse consequences have occurred, despite the existence of established screening tools. As a potential method of early detection, we examined whether readily available and routinely measured clinical biochemical diagnostic test data could predict poor nutritional status. We combined 2008–2017 data of 1518 free-living individuals ≥50 years from the United Kingdom National Diet and Nutrition Survey (NDNS) and used logistic regression to determine associations between routine biochemical diagnostic test data, micronutrient deficiency biomarkers, and established malnutrition indicators (components of screening tools) in a three-step validation process. A prediction model was created to determine how effectively routine biochemical diagnostic tests and established malnutrition indicators predicted poor nutritional status (defined by ≥1 micronutrient deficiency in blood of vitamins B_6_, B_12_ and C; selenium; or zinc). Significant predictors of poor nutritional status were low concentrations of total cholesterol, haemoglobin, HbA1c, ferritin and vitamin D status, and high concentrations of C-reactive protein; except for HbA1c, these were also associated with established malnutrition indicators. Additional validation was provided by the significant association of established malnutrition indicators (low protein, fruit/vegetable and fluid intake) with biochemically defined poor nutritional status. The prediction model (including biochemical tests, established malnutrition indicators and covariates) showed an AUC of 0.79 (95% CI: 0.76–0.81), sensitivity of 66.0% and specificity of 78.1%. Clinical routine biochemical diagnostic test data have the potential to facilitate early detection of malnutrition risk in free-living older populations. However, further validation in different settings and against established malnutrition screening tools is warranted.

## 1. Introduction

Malnutrition (or “undernutrition”) is a major cause of poor health in older age and is associated with loss of body weight and skeletal muscle mass. As such, it is linked with sarcopenia [1], the loss of skeletal muscle mass and function with age [2] and reduced immuno-resilience [3], as well as with frailty, falls and fractures [4,5,6] and premature mortality [7,8]. Malnutrition is estimated to affect more than 3 million individuals in the United Kingdom (UK), with more than one-third over the age of 65 years [9]. Malnutrition is not only highly prevalent in hospitals [10], where, at admission, about 30% of patients are at risk of malnutrition [11], but is also frequently present prior to hospital admission in community-dwelling older people living at home or in residential care [12,13]. Detecting malnutrition in the community is therefore of great importance. Health and social care costs associated with the detrimental effects of malnutrition are estimated to be £23.5 billion in the UK, with costs for malnourished patients estimated to be three times higher than for others [14], due to increased visits to primary care, hospital admissions and duration of hospitalisation [8,14].

Malnutrition can be a consequence of inadequate intake of macro- or micronutrients, leading to undernutrition, protein-energy malnutrition (PEM) and/or micronutrient deficiency, resulting in recognisable physiological, metabolic and clinical symptoms. Micronutrient deficiencies alone are recognised as contributing to substantial health consequences in older adults, including frailty and sarcopenia [5,15,16,17,18,19,20,21,22]. Other than an inadequate dietary intake, impairments in nutrient absorption, transport and utilization, accompanied by clinical conditions, may also contribute to the development of malnutrition that also includes micronutrient deficiency [23]. The physiological consequences of malnutrition in older adults are often difficult to reverse.

The detrimental health effects of malnutrition on individuals as well as its significant health and social care costs reinforce the need for early detection and intervention to address malnutrition. However, despite the use of “established malnutrition screening tools” (e.g., Mini Nutritional Assessment (MNA) [24], Subjective Global Assessment (SGA) [25] and Nutritional Risk Screening (NRS 2002) [26]) in community, secondary and residential care settings, malnutrition in older adults is often underdiagnosed until a critical state is reached. Moreover, these tools are used inconsistently, and nutritional intervention or treatment is not always provided [14].

In order to provide a more readily available tool for early malnutrition screening and detection of at-risk free-living individuals in the community, we investigated the potential of blood measurement data from clinical “routine biochemical diagnostic tests” to predict poor nutritional status. Routine biochemical diagnostic tests are readily available from health checks in middle and older aged individuals in primary and secondary care in many healthcare systems in Europe and the United States (US). Since these biochemical tests are performed routinely, whereas malnutrition screening is not, these biochemicals may be useful to alert to the presence of under- or malnutrition. Routinely measured biochemical diagnostic tests include measurements of cholesterol, haemoglobin A1C (HbA1c), creatinine and Estimated Glomerular Filtration Rate (eGFR), which are typically used to determine risk and treatment for conditions, such as cardiovascular disease, diabetes or renal failure, and are predictive of morbidity and mortality [27,28,29]. Certain routinely used biochemical tests have also been shown to be associated with the risk of micronutrient insufficiency [30,31] and with established malnutrition screening tools in a secondary care setting [32]. However, they have not yet been tested in a community setting. To our knowledge there are no previous studies that investigated both cut-off points for biochemical tests that are used in clinical practice for alerting clinicians to risk or treatment of disease, alongside cut-off points used for alerting clinicians specifically to risk of malnutrition.

Our analysis used data from the National Diet and Nutrition Survey (NDNS), which measures food consumption, nutrient intakes and nutritional status of a representative sample of people living in private households in the UK. The NDNS includes data on “established malnutrition indicators”, i.e., individual components included in established malnutrition screening tools and risk factors for malnutrition (e.g., low protein intake, poor appetite and polypharmacy [33]), but did not administer complete malnutrition screening tools within the survey. Moreover, the NDNS measured “biomarkers of micronutrient deficiency” in blood, which are associated with clinically recognisable symptoms and morbidities, as well as being accepted measures of dietary intake [5,15,16,17,18,19,20,21,22].

The current study therefore investigated whether routine biochemical diagnostic tests that are routinely measured in clinical practice (e.g., HbA1c, cholesterol and eGFR) have the potential to detect and alert clinicians to the presence of an increased risk of malnutrition in free-living individuals of 50 years or older (Figure 1). We validated these routine biochemical diagnostic tests firstly against directly measured micronutrient deficiency biomarkers not routinely measured in clinical practice, and secondly against established malnutrition indicators. In a third step, we explored relationships between micronutrient deficiency biomarkers and established malnutrition indicators. Finally, we generated an exploratory risk prediction model to test the degree of prediction of a poor nutritional status, assessed by the presence of micronutrient deficiency, by routine biochemical diagnostic tests and established malnutrition indicators.

## 2. Materials and Methods

### 2.1. Study Design and Population

The NDNS Rolling Programme is a cross-sectional annual survey designed to assess the diet, nutrient intake and nutritional status of a representative sample of people living in private households in the UK. The NDNS rolling programme started in 2008/09 and is currently in its eleventh year. A detailed description of the NDNS methodology is published elsewhere [34,35,36,37]. In brief, a sample of ~1000 people was selected each year from randomly sampled addresses from all UK postcodes, of which 500 were adults and 500 were children. Demographic data, a 4-day food diary, physical measurements and a blood and urine sample were collected by a trained interviewer and nurse. From the blood sample, plasma or serum concentrations of a range of micronutrients relevant to public health and for monitoring dietary intake were measured. The NDNS Rolling Programme was conducted according to the Declaration of Helsinki. Ethical approval was obtained from the Oxfordshire A Research Ethics Committee (Reference No. 07/H0604/113) for years 2008/09 to 2012/13 and from the Cambridge South NRES Committee (Reference No. 13/EE/0016) for years 2013/14 to 2016/17. Informed consent was obtained from all subjects.

For our study, NDNS data from years one to nine (2008/09 to 2016/17) were combined to generate a dataset spanning all years for analysis of individuals aged 50 years or older living in private households (Figure 2). A total of 1518 subjects with valid laboratory measurements for plasma vitamin C, vitamin B6 Pyridoxal 5′-Phospate (PLP), selenium, zinc and serum vitamin B12 were included. We have used available case analysis, where cases were dropped if values were missing for that specific measurement, meaning the analyses were based on different subsets of the data. Relevant numbers are specified in the tables.

### 2.2. Study Population Measurements

#### 2.2.1. Subject’s Characteristics Data Collection

The following demographic and lifestyle data were obtained from a detailed interview: sex (men, women), age category (50–59 years, 60–69 years, ≥70 years), ethnicity (white British, other), region (England—North, England—Central/Midlands, England—South (including London), Scotland, Wales, Northern Ireland), qualification (secondary education or less, further education, higher education and other), marital status (single (never married), married or legally recognised civil partnership, divorced or widowed), self-assessed general health (good, fair, bad), presence of longstanding illness (yes, no), any dietary supplement use in the past year (yes, no), appetite (good, average, poor) and whether subjects had any of their own teeth (yes, no). Cigarette smoking status (never, former, current) was obtained through a self-completion questionnaire. Anthropometric measurements, height and weight, were measured by using a portable stadiometer and weighing scales. Body mass index (BMI) was calculated as weight (kg)/height squared (m^2^). During the nurse visit, a blood sample was collected and subjects were asked the number of medicines they used (no medication, 1–4 medicines or 5 or more medicines (defined as polypharmacy [38])). Further details about data collection and categorization can be found in Appendix A Appendix A.

#### 2.2.2. Dietary Data Collection

Dietary data collection and analyses used in the NDNS have been described in detail elsewhere [39,40,41,42]. In brief, four-day food diaries were collected for each participant, where participants were asked to keep a record of everything eaten or drunk over four consecutive days. During the first interviewer visit an explanation was given of the dietary-data collection methods, including an explanation on describing details of food and drink and portion sizes. The diaries included pictures of different portion sizes to support accurate recording. Completed food diaries were coded, processed and quality checked. Nutritional information including protein and energy intake (g/day) and the number of portions of fruit and vegetable consumed per day was calculated by using food-composition data of Public Health England’s NDNS Nutrient Databank. One portion of fruit and vegetables was defined as 80 g; fruit juice (from all sources) and pulses (including baked beans) were included in the calculation up to a maximum of one portion per day each, where one portion fruit juice was defined as 150 g. For our study, fluid intake was calculated by the sum in grams of the following fluids: fruit juice (including smoothies), soft drinks (low calorie), soft drinks (not low calorie), tea, coffee, water, dairy, beer, lager, cider and perry.

#### 2.2.3. Blood Sample Collection and Biochemical Analyses

Blood samples were drawn after an overnight fast, except for people with diabetes not willing to fast, in which case a non-fasting blood sample was taken. Blood samples were collected by venepuncture and analysed for a wide range of biochemical markers, conducted at either the Medical Research Council Elsie Widdowson Laboratory (MRC EWL) or the Immunology and Biochemistry Laboratory at Addenbrooke’s Hospital in Cambridge. Full details of blood sample collection and analyses used in the NDNS are described elsewhere [37]. In general, blood sample procedures remained constant over NDNS years 1–9; however, when changes were needed, crossover studies were carried out to ensure the comparability of results over time [37]. Micronutrient assays relevant to the outcome of the present study are described in Appendix A Appendix A.

### 2.3. Routine Biochemical Diagnostic Tests and Cut-Off Points for Inadequate Status

We defined “routine biochemical diagnostic tests” as any biochemical or haematology test frequently measured in clinical practice to determine risk of particular diseases. The following routine biochemical diagnostic tests were selected for analyses: cardiovascular disease (serum total cholesterol, serum triglycerides, high-density lipoproteins (HDL), low-density lipoproteins (LDL)), anaemia and iron status (haemoglobin, haematocrit, mean cell volume, plasma ferritin), diabetes (glycated haemoglobin HbA1c), inflammatory status or infection (lymphocyte count, white blood cell count, serum C-reactive protein (CRP)), renal function (eGFR, plasma creatinine) and risk of bone disease (plasma 25-hydroxy vitamin D (vitamin D status)). The selection of routinely measured diagnostic tests as potential predictors of poor nutritional status was mainly based on the availability in the dataset and additionally on measures used in the systematic review of Zhang et al. (2017) [32]. Measures of vitamin D and iron status were included as routine biochemical diagnostic tests since they are frequently measured in older populations due to the growing recognition of the high prevalence of inadequate status in the general population. Values for eGFR were derived by the Chronic Kidney Disease Epidemiology Collaboration (CKD-EPI) equation [43].

A majority of the cut-off points we have used for inadequate status are those widely used by clinicians to assist in determining whether individuals are at risk of disease or poor health status. As most of these cut-off points are not developed to specifically identify malnutrition, a more extensive literature review was conducted on suggested cut-off points specifically for malnutrition. For the routine biochemical diagnostic tests, the lower cut-off points were mostly derived from established reference ranges used in clinical practice [44]. Low cholesterol was based on the cut-off point for hypocholesterolaemia, as described in previous studies [45,46]. The cut-off value for eGFR was based on the grading of renal function, according to the National Institute for Health and Clinical Excellence (NICE) guidelines [47], and the vitamin D deficiency cut-off point was derived from the UK Scientific Advisory Committee on Nutrition (SACN) [48]. No official cut-off point was available for low HbA1c. HbA1c concentrations < 4.0% were associated with an increased risk of all-cause mortality in a general population without diabetes [28], and HbA1c concentrations of below approximately 6.5% were associated with increased mortality risk in diabetic populations [49]. As only a few subjects had HbA1c concentrations below 4.0% and diabetic subjects were included in the current study population, a cut-off point of 5.0% was chosen.

### 2.4. Micronutrient Deficiency Biomarkers and Cut-Off Points for Inadequate Status

In this study a poor nutritional status was defined as the presence of at least one inadequate blood concentration out of five micronutrients not routinely measured in clinical practice (plasma vitamin C, plasma vitamin B6 PLP, serum vitamin B12, plasma selenium and plasma zinc). Ideally, the selection of these micronutrients would have been based on full data of established malnutrition screening tools; however, these data were not available. Since a previous systematic review found associations between malnutrition and BMI, haemoglobin, and total cholesterol in a clinical setting [32], these three proposed malnutrition indicators were used for micronutrient selection. From the top ten most prevalent micronutrient inadequacies in the total study population (*n* = 3284) (vitamin B2 (erythrocyte glutathione reductase activation coefficient (EGRAC)), vitamin B6 PLP, selenium, vitamin B1 (erythrocyte transketolase activation coefficient (ETKAC)), zinc, red cell folate, vitamin B12, vitamin C, retinol and α-tocopherol), those found to be individually associated with low BMI, haemoglobin and/or total cholesterol in the study population with valid micronutrient measurements (*n* = 1218, data not shown) (adjusted for age categories and sex) were selected. Retinol and α-tocopherol were already excluded because only a low proportion of the population was deficient in these micronutrients (~0.4%). In addition to the absence of associations of vitamin B2, vitamin B1 and red cell folate with our three chosen indicators, other reasons for exclusion were uncertainty about the clinical relevance of some cut-off points (e.g., an EGRAC > 1.3 for vitamin B2 [34,35]), and fewer subjects who had valid measurements for the eight remaining micronutrients. The selected five micronutrients were not highly correlated with each other, indicating they reflect different aspects of the diet (data not shown).

Cut-off points of deficiency are defined as concentrations below reference ranges which relate to being at risk of adverse health outcomes. These were obtained from reports of various public health institutes and organisations, including the European Food Safety Authority (EFSA), Norfolk and Norwich University Hospital (NNUH) and World Health Organization (WHO). For vitamin B12, vitamin C and selenium cut-off points defining deficiency were chosen. For vitamin B6 PLP and zinc it was less clear whether the chosen cut-off points defined deficient or suboptimal status, because of the lack of consensus about cut-off points for deficient and suboptimal micronutrient status.

### 2.5. Established Malnutrition Indicators and Cut-Off Points for Inadequate Status

Individual components of established malnutrition screening tools and malnutrition risk factors were selected and defined as “established malnutrition indicators”. The selection of established malnutrition indicators as potential predictors of a poor nutritional status was based on previous research studying risk factors for malnutrition in older adults [33,50,51,52], as well as on the availability of these risk factors for malnutrition in the NDNS dataset. These included BMI, protein intake, energy intake, fruit and vegetable intake, fluid intake, marital status, self-assessed general health, the presence of one or more long-standing illness(es), polypharmacy, dietary supplement use, the presence of own teeth and appetite.

Cut-off points for various established malnutrition indicators are described above. Low BMI was defined as <20 kg/m^2^ when age was <70 years and <22 kg/m^2^ when age was ≥70 years, using the Global Leadership Initiative on Malnutrition (GLIM) criteria [53]. Low protein intake was defined as a dietary protein intake (g) below the Reference Nutrient Intake (RNI), which is set at 0.75 g protein per kilogram body weight per day in adults according to the UK Dietary Reference Value (DRV) [54]. Low energy intake was defined as an energy intake (kcal) below the Estimated Average Requirement (EAR) reported by the SACN, which were adjusted to sex and age range [55]. The energy requirement could not be adjusted to physical activity level because of unavailable data. A low fruit and vegetable intake was defined as either <5 portions/day, in accordance with the Eatwell Guide [56], or <2 portions/day, based on the question on fruit or vegetable consumption in the MNA. Low fluid intake was defined as <1600 mL/day for women and <2000 mL/day for men derived from the EFSA guidelines for water intake through drinking water and beverage consumption (80% of water intake) [57]. Additional cut-off points for fluid intake were <1250 mL/day (<5 cups) and <750 mL/day (<3 cups), based on the question on fluid consumption in the MNA.

### 2.6. Statistical Analyses

Statistical analyses of the data were performed by using Stata (release 16.0; StataCorp LLC, College Station, TX, USA) and IBM SPSS Statistics (version 25.0, IBM Corp., Armonk, NY, USA). Statistical significance was based on two-sided values of *p* < 0.05. Continuous variables are expressed as mean ± standard deviation (SD) and categorical variables are expressed as frequencies and percentages (%). Descriptive statistics for characteristics of the study population were calculated, stratified by micronutrient deficiency biomarker status. Analyses to test differences between micronutrient deficiency biomarker status groups were performed by ANOVA for continuous variables and Pearson’s Chi-Square tests for categorical variables. Binary variables were created for each of the selected (micro)nutrients and routine biochemical diagnostic tests, with the categories “deficient” or “sufficient” (“inadequate” or “adequate”), according to the defined cut-off points. Where measurements for the routine biochemical diagnostic tests were invalid, subjects were assigned to a “missing data” category for that specific routine biochemical to prevent the loss of data of other variables in the model.

A binary outcome variable defined as “at least one micronutrient deficiency vs. no micronutrient deficiencies” was generated. There were no subjects deficient in more than two micronutrients. Prior to multivariable logistic regression analyses, multicollinearity between continuous blood concentrations of the routine biochemical diagnostic tests as well as between the established malnutrition indicators was explored by detecting Pearson Correlation Coefficients of greater than 0.7 in a correlation matrix as well as variance inflation factors (VIFs). Based on these criteria, haematocrit (haematocrit and haemoglobin: r > 0.9), LDL (LDL and total cholesterol: r > 0.9) and creatinine (creatinine and eGFR: r > 0.7) were excluded from multivariable analyses. Results from logistic regression analyses were expressed as crude and adjusted odds ratios (OR) with 95% confidence intervals. The analysis was conducted in four stages:

Firstly, (Stage 1a) to examine associations between low concentrations of routine biochemical diagnostic tests and micronutrient deficiency (as defined by the binary outcome variable described above), separate logistic regression analyses were conducted for each biochemical. These analyses were then repeated, adjusted for relevant covariates (sex, age categories, ethnic group, region, qualification and smoking status). Thereafter, (Stage 1b) we tested a model that included all routine biochemical diagnostic tests together, adjusted for the covariates as before.

Secondly, (Stage 2) as further validation the routine biochemical diagnostic tests associated with at least one micronutrient deficiency in Stage 1b were tested in logistic regression analyses against established malnutrition indicators, adjusted for the covariates.

Thirdly, (Stage 3) to examine the associations between established malnutrition indicators and the presence of micronutrient deficiency, separate logistic regression analyses were tested for each established malnutrition indicator, adjusted for the covariates.

The next and last stage (Stage 4) was the creation of a model predicting a poor nutritional status (the presence of at least one micronutrient deficiency). The routine biochemical diagnostic tests, covariates, and established malnutrition indicators were entered in one multivariable logistic regression model. A manual stepwise backward elimination was then performed based on *p*-values ≤ 0.05 for variable selection. Protein intake and energy intake were excluded before variable selection, because of the scarcity of dietary intake data collection in primary-care settings. In addition, based on the analysis of Stage 3, only a fruit and vegetable intake < 2 portions/day and fluid intake < 1600 mL/day (women) or <2000 mL/day (men) were included for variable selection, because these were significantly related to the presence of at least one micronutrient deficiency and had the greatest OR. In order to base the variable selection on adjusted odds ratio’s, the covariates (sex, age categories, ethnic group, region, qualification and smoking status) were locked into the multivariable logistic regression model.

The predictive performance of the final model was evaluated: Overall performance was judged by the Brier score (where a score of 0 indicates perfect agreement of the prediction on the outcomes, and 1 indicates perfect disagreement). The discriminative ability was judged by receiver-operating characteristic (ROC) curves and the corresponding area under the curve (AUC), also known as the “concordance” (C)-statistic. AUC values of 0.5 indicate a model that predicts no better than chance, while values of 1 indicate perfect discrimination. Accuracy (calibration) was judged by the Hosmer–Lemeshow goodness-of-fit test. The sensitivity and specificity of the final model were shown to describe the model’s screening ability for a poor nutritional status relative to the reference standard (micronutrient status measured in blood). Positive and negative predictive values were also calculated.

## 3. Results

### 3.1. Study Population Characteristics

Characteristics of the study population are presented in Table 1, stratified by micronutrient deficiency biomarker status. The present study included 1518 subjects, of whom 789 (52.0%) were not deficient in any of the selected micronutrients (vitamin B6, vitamin B12, vitamin C, selenium, and zinc). A total of 729 (48.0%) were deficient in at least one out of five micronutrients. No subjects were deficient in more than two micronutrients. The total study population had a mean age of 64 (SD 9.7) years and consisted of more than half of women (57.2%). The majority was white British, married and had a good self-rated health, with 13.2% of the study population currently smoking. More than half of subjects had a longstanding illness (56.1%), with approximately two-thirds of the population using prescribed medication. In addition, more than one-third used any dietary supplementation at least once in the past 12 months. The mean BMI of the total study population was 28.2 (SD 5.0), with the mean BMI of men (28.6 ± 4.4) slightly higher than the mean BMI of women (28.0 ± 5.5). A total of 3.5% of subjects had a BMI below 20 (age < 70 years) or below 22 (age ≥ 70 years), with 2.4% of men and 4.7% of women having a low BMI. Furthermore, almost a quarter of the population had a protein intake below the RNI (22.5%), the majority reported an energy intake below the EAR (84.9%), 14.6% consumed less than two portions of fruit and vegetables per day, and more than half did not meet the EFSA guidelines for fluid intake (59.8%). Study population characteristics further stratified by no, one or two micronutrient deficiencies are presented in Appendix A Appendix A.

Most characteristics differed between the absence or presence of micronutrient deficiencies, except for sex, ethnicity, and energy intake below the EAR. Subjects with at least one micronutrient deficiency tended to be in the highest age category, had a lower qualification, and were more often divorced or widowed, and a smoker. In terms of health, this group had a higher mean BMI, as well as more subjects with a longstanding illness, polypharmacy, and a worse self-assessed general health, with fewer subjects taking any type of dietary supplements. In terms of nutritional intake, the prevalence of protein intake below the RNI, a low fruit and vegetable intake, and a fluid intake below 1600 mL/day for women and 2000 mL/day for men, were all higher in those with at least one micronutrient deficiency.

For comparison, the characteristics of subjects included for analysis (*n* = 1518, 46.2%) and the total population (*n* = 3284, 100%) are shown in Appendix A Appendix A. Although not statistically tested, no major differences between the subjects included for analyses and the total study population are evident.

### 3.2. Descriptives of Micronutrient Deficiency Biomarkers and Routine Biochemical Diagnostic Tests

Mean micronutrient blood concentrations were all above the selected cut-off points for inadequate status (Table 2). Approximately one-third of the study population had an inadequate vitamin B6 status, followed by a quarter of the study population with inadequate selenium concentrations. Fewer subjects were inadequate in zinc, vitamin B12 or vitamin C. The proportion of subjects with inadequate status was approximately equal between men and women for each micronutrient.

In general, a relatively small proportion of the study population had values below the lower cut-off point of the reference range or above (MCV and CRP), according to their blood concentrations of the selected routine biochemical diagnostic tests (Table 3). Especially for triglycerides, HbA1c, and creatinine only few subjects had low concentrations (between 0.9% and 2.4%). Mean concentrations of the routine biochemicals differed significantly between subjects not deficient in any of the micronutrients and subjects deficient in at least one of the micronutrients, except for lymphocyte count, mean cell volume, ferritin, and creatinine (only for women). Mean concentrations of triglyceride, HbA1c, white blood cell count, CRP and creatinine were higher in deficient subjects compared to non-deficient subjects.

### 3.3. Associations between Routine Biochemical Diagnostic Tests and Micronutrient Deficiency (Stage 1)

Table 4 presents crude and adjusted OR’s of univariate analyses, where each of the routine biochemicals were tested individually in both unadjusted models and after adjusting for relevant covariates. A total of 11 out of 15 tested biochemicals were significantly associated with the presence of at least one micronutrient deficiency after adjustment for relevant covariates. It should be noted the confidence intervals for the OR of creatinine with at least one micronutrient deficiency were wide. Adjusted OR’s in multivariable analyses are also presented in Table 4, where all selected routine biochemicals were tested together, and adjusted for relevant covariates. Low total cholesterol, haemoglobin, ferritin, HbA1c, eGFR, vitamin D status and high CRP were all significantly associated with the presence of at least one micronutrient deficiency. The adjusted OR’s suggest a high CRP is the strongest predictor of micronutrient deficiencies.

### 3.4. Associations between Routine Biochemical Diagnostic Tests and Established Malnutrition Indicators (Stage 2)

Unadjusted and adjusted associations between routine biochemical diagnostic tests and established malnutrition indicators are shown in Table 5. In brief, except for a low HbA1c, all tested routine biochemicals were significantly associated with at least one of the established malnutrition indicators. Low total cholesterol, haemoglobin, eGFR, vitamin D status, and high CRP concentrations were significantly associated with a protein intake below the RNI in adjusted analyses (OR’s ranging from 1.74 to 2.35), whereas none of the tested biochemicals were related to energy intake below the EAR. A low ferritin concentration was positively associated with a low BMI. In addition, subjects with low haemoglobin or low vitamin D concentrations were 1.5 to 2 times as likely to consume less than two portions of fruit and vegetables per day and to have a fluid intake below 1600 mL/day for women and 2000 mL/day for men. A low HbA1c was not related to any of the selected established malnutrition indicators.

### 3.5. Associations between Established Malnutrition Indicators and Micronutrient Deficiency (Stage 3)

Table 6 presents adjusted OR’s of univariate analyses, where established malnutrition indicators were tested individually against the presence of at least one micronutrient deficiency, adjusted for relevant covariates. Low protein intake was significantly associated with the presence of at least one micronutrient deficiency, as well as both a protein intake below the RNI and an energy intake below the EAR. Energy intake below the EAR on its own was not significantly associated with the presence of at least one micronutrient deficiency. At a more extreme energy intake of lower than 1800 kcal per day, an association with at least one micronutrient deficiency was found. A fruit and vegetable intake below five portions per day and below two portions per day were both significantly associated with the presence of at least one micronutrient deficiency, as well as a fluid intake below the EFSA guidelines. For fluid intake below the MNA cut-off points no associations were found. Likewise, low BMI was not significantly associated with at least one micronutrient deficiency.

### 3.6. Prediction of a Poor Nutritional Status (Micronutrient Deficiency) (Stage 4)

Table 7 shows the final model predicting a poor nutritional status (the presence of at least one micronutrient deficiency versus no micronutrient deficiencies). Routine biochemical diagnostic tests predicting micronutrient deficiency were low total cholesterol, low haemoglobin, low ferritin, low HbA1c, low vitamin D status and a high CRP. Subjects with CRP concentrations above 10 mg/L were more likely to have one or two micronutrient deficiencies. Moreover, subjects with haemoglobin concentrations below 13 g/dL for men and 12 g/dL for women, ferritin concentrations below 23 µg/L, HbA1c concentrations below 5.0%, or vitamin D concentrations below 25 nmol/L were approximately two to three times as likely to have at least one micronutrient deficiency. Subjects with total cholesterol concentrations below 4.1 mmol/L were slightly more likely to have at least one micronutrient deficiency.

Established malnutrition indicators predicting at least one micronutrient deficiency were polypharmacy as opposed to not using any medication, having a poor appetite, a bad self-assessed general health and less than 2 portions of fruit and vegetable intake per day. Subjects using any type of dietary supplement in the past year were less likely to have a micronutrient deficiency. Covariates predicting at least one micronutrient deficiency were a higher age category and living in Wales as opposed to living in Central England or the midlands. Furthermore, current smokers were three times more likely to have multiple micronutrient deficiencies than never smokers. The remaining routine biochemical diagnostic tests as well as BMI, fluid intake, sex, ethnic group, qualification, marital status, the presence of a longstanding illness and having any of your own teeth were not found to be predictors of at least one micronutrient deficiency.

The area under the ROC curves between five different models (Model 1, “routine biochemical diagnostic tests”; Model 2, “established malnutrition indicators”; Model 3, “routine biochemical diagnostic tests and covariates”; Model 4, “established malnutrition indicators and covariates”; and Model 5, “routine biochemical diagnostic tests, established malnutrition indicators and covariates”) differed significantly (*p* ≤ 0.001) (Figure 3). Model 1, consisting of only the routine biochemical diagnostic tests, showed average discrimination (0.67 (95% CI: 0.64–0.69)). Model 2, consisting of only the established malnutrition indicators, showed fair discrimination (0.71 (95% CI: 0.68–0.74)). The addition of covariates to these models increased the AUC significantly, resulting in comparable discrimination of Models 3 and 4 (0.76 (95% CI: 0.73–0.78) vs. 0.75 (95% CI: 0.72–0.77), respectively). The final model (Model 5), including routine biochemical diagnostic tests, established malnutrition indicators and covariates, showed the best discrimination (0.79 (95% CI: 0.76–0.81)). The Hosmer–Lemeshow goodness-of-fit test (*p* = 0.513) shows there is no evidence of poor fit of the final model and the Brier score (0.19) indicates good agreement of the obtained predictions on the outcomes. Sensitivity and specificity values of the model were 66.0% and 78.1% respectively, and positive and negative predictive values were 73.6% and 71.3%.

## 4. Discussion

This study has demonstrated that biochemical diagnostic tests routinely used in clinical practice for determining risk of cardiovascular disease, anaemia, diabetes, renal failure, or vitamin D deficiency, could also potentially facilitate early screening of poor nutritional status in free-living older individuals. Our multiple validation of these diagnostic tests, firstly against micronutrient deficiency biomarkers in individuals with one or more independently measured deficiencies (of plasma vitamin C, selenium, zinc, vitamin B6 or serum vitamin B12), and secondly against risk factors of malnutrition and established malnutrition indicators that comprise components of established malnutrition screening tools, supports further research into their use as screening tools to alert for poor nutritional status and malnutrition.

The biochemical tests, total cholesterol, haemoglobin, ferritin, 25-hydroxy vitamin D, and CRP were predictors of poor nutritional status (defined by at least one micronutrient deficiency); apart from HbA1c, they were also associated with several of the established malnutrition indicators that comprise components of established screening tools. The third stage of our validation, showing associations between established malnutrition indicators and micronutrient deficiency, not only confirms the links between malnutrition and micronutrient deficiency, but also our findings of the associations between the biochemical diagnostic tests and micronutrient deficiency.

Although malnutrition screening and treatment have proven value for health, there are logistical difficulties in performing regular malnutrition screening. The biochemical diagnostic tests we examined are widely used in primary and secondary care in many healthcare systems in Europe and the US to assist in diagnosis or treatment of clinical conditions. They provide the benefit of being readily and routinely available in clinical care to potentially highlight middle and older aged individuals at risk of poor nutritional status and malnutrition. Measures of vitamin D deficiency and anaemia are also measured frequently in clinical practice and, thus, may similarly be useful; therefore, they were included in our models. A strength of our study is the use of micronutrient deficiency biomarkers, not routinely measured in clinical practice, for validation of the biochemical diagnostic tests and established malnutrition indicators. Micronutrient deficiencies are highly relevant as they contribute to adverse health consequences in older adults [5,15,16,17,18,19,20,21,22] and are reflective of poor dietary quality. Moreover, micronutrient biomarkers are established measures for determining micronutrient deficiency, and cut-off points known to cause physiological and metabolic symptoms were used to determine deficiency. A further advantage is the absence of “dietary reporting bias”, in comparison to self-reported dietary intake [31,63]. Our findings are mostly generalizable to other community populations since the study population was derived from a randomly recruited population sample, and no substantial differences between our study population and the larger available NDNS population were found. Generalizability to other ethnic groups may be limited as ≥95% of the NDNS population was white British (87.1% of UK individuals identify as of white ethnicity [64]), and previous studies show non-white older adults are at increased risk of malnutrition [65,66].

The high prevalence of older individuals with one or more deficiencies of vitamin C, selenium, zinc, or vitamins B6 or B12 (48%), despite excluding vitamin D deficiency and anaemia, indicates a high proportion of the population is at risk of substantial adverse health consequences, comparable to those of PEM, and should be given more attention.

The lack of previous research in this area highlights the novelty of this study. Previous reviews have proposed low blood concentrations of haemoglobin and total cholesterol as biochemical indicators for malnutrition risk in older adults from various clinical settings [32,45]. Our results support and extend these findings to older free-living individuals. In contrast to a previous study [32], we found elevated CRP predicted micronutrient deficiency and low protein intake. A possible reason for this discrepancy between studies is that mean CRP concentration in our study was substantially lower, perhaps giving us a better scope to identify differences. However, the utility of CRP measurements may be limited, particularly in populations less healthy than the NDNS, as CRP is very responsive to acute inflammation [32,67]. Nevertheless, CRP may indicate whether serum protein concentrations of albumin are reduced because of inflammatory processes or malnutrition [67]. Our data showed that vitamin D deficiency predicted other micronutrient deficiencies. This is supported by previous evidence of vitamin D serum concentrations decreasing significantly with malnutrition risk [68]. It should be noted that vitamin D deficiency does not solely result from inadequate dietary intake, but also from inadequate sun exposure. Low concentrations of HbA1c have not been researched extensively in relation to malnutrition, as typically high concentrations of HbA1c are used to detect and diagnose diabetes. Previous research suggests that those with lower HbA1c levels had a higher prevalence of malnutrition, weight loss and other comorbidities [49]. Still, the association between lower HbA1c and micronutrient deficiency in this study should be interpreted with caution, as numbers of subjects with HbA1c < 5.0% were small and no strict lower cut-off point was available in our search of the literature.

Although low BMI is included as a key indicator for malnutrition in many nutritional assessment tools, in this study a low BMI was not associated with micronutrient deficiencies. Similarly, another study showed BMI < 23.9 kg/m^2^ was not associated with low micronutrient status in an elderly UK population [69]. As only 3.7% of our study population had a low BMI, the statistical power to observe relationships may have been lacking. To enable early identification of malnutrition in free-living older adults, it has been suggested that the use of a higher BMI cut-off point may be more useful [32].

### 4.1. Limitations

The lack of a direct comparison with complete established malnutrition screening tools (e.g., MNA) limits our conclusions, as does the unavailability of measurements of body fat or muscle mass (e.g., (triceps) skinfold thickness and mid-upper arm circumference [67]). Blood concentrations of routine biochemicals and micronutrients were measured at a single time-point and may have been affected by acute conditions, including inflammation, in particular plasma levels of zinc, selenium and vitamin B6 PLP [70,71,72]. Furthermore, as we studied a highly prevalent outcome, ORs may have been overestimated in our models and, thus, should not be interpreted as risk ratios. Our final predictive model showed fair sensitivity and good specificity (66.0% and 78.1%, respectively), indicating the correct identification of individuals with micronutrient deficiency is difficult and a proportion may be missed. Another limitation is the absence of a separate training and test dataset, due to the limited number of subjects available; therefore, further testing of our model in other study populations is necessary. Finally, the cross-sectional design of this study limits proof of causal relationships.

### 4.2. Recommendations

Although we do not consider our proposed routine biochemical diagnostic tests as a replacement for established malnutrition screening tools, a range of these biochemical tests have potential for providing a simpler means for early detection of poor nutritional status risk and the associated adverse health outcomes, and to alert physicians to the potential need of further investigations into and treatment for malnutrition. Further testing of these biochemical tests and readily available malnutrition risk factors (e.g., age, smoking habits and polypharmacy) in larger study populations in different settings, with further validation against established malnutrition screening tools and measures of muscle mass, is warranted. Furthermore, to improve sensitivity and specificity of (pre-)malnutrition identification with biochemical tests, cut-off points might need adaptation. As better predictive performance resulted from combining the most predictive biochemical tests with established malnutrition indicators, further testing of optimal combinations of routinely collected data is recommended.

## 5. Conclusions

To our knowledge, this is the first study to have used a multiple-step validation process to examining direct biochemical measures of micronutrient deficiency and components of malnutrition screening tools, including protein intake, to investigate the use of routinely measured biochemical diagnostic test data in order to identify poor nutritional status and malnutrition. Although our results suggest that routine biochemical diagnostic tests have the potential to facilitate early malnutrition screening in free-living middle-aged and older-age populations, further assessment and refining in larger populations and comparison with established malnutrition screening tools are required.

## Figures and Tables

**Figure 1 nutrients-13-01883-f001:**
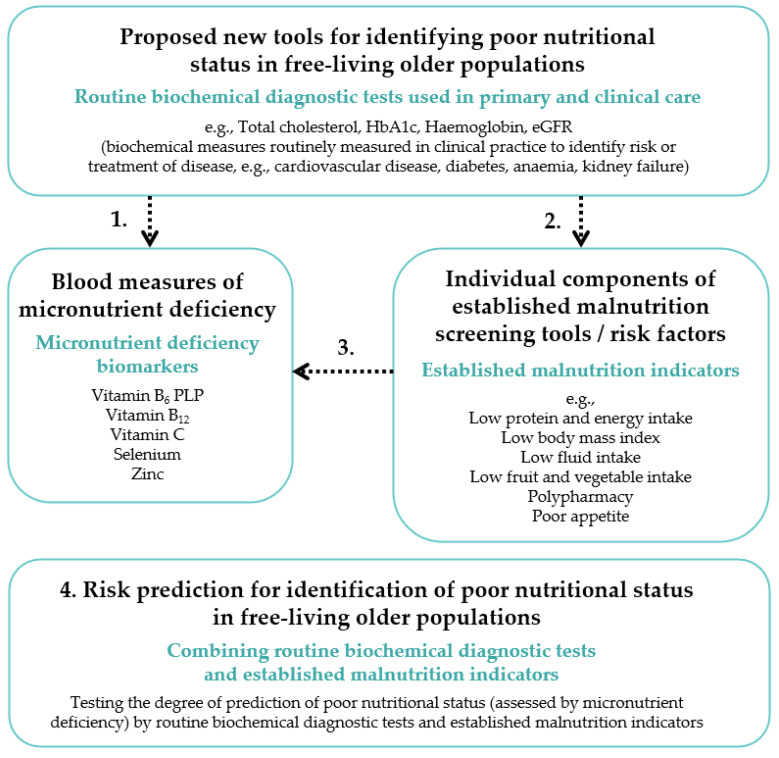
Schematic overview of the concepts used in the investigation of routinely measured diagnostic tests as potential predictors of poor nutritional status/malnutrition risk in free-living older populations. For analyses, a poor nutritional status was defined as the presence of ≥1 micronutrient deficiency. Stage 1: Micronutrient deficiency biomarkers used as validation for routine biochemical diagnostic tests used in primary and clinical care. Stage 2: Individual components of established malnutrition screening tools/risk factors (established malnutrition indicators) used as further validation for routine biochemical diagnostic tests. Stage 3: Micronutrient deficiency biomarkers used as validation for established malnutrition indicators. Stage 4: Routine biochemical diagnostic tests as potential predictors of poor nutritional status risk in free-living older populations.

**Figure 2 nutrients-13-01883-f002:**
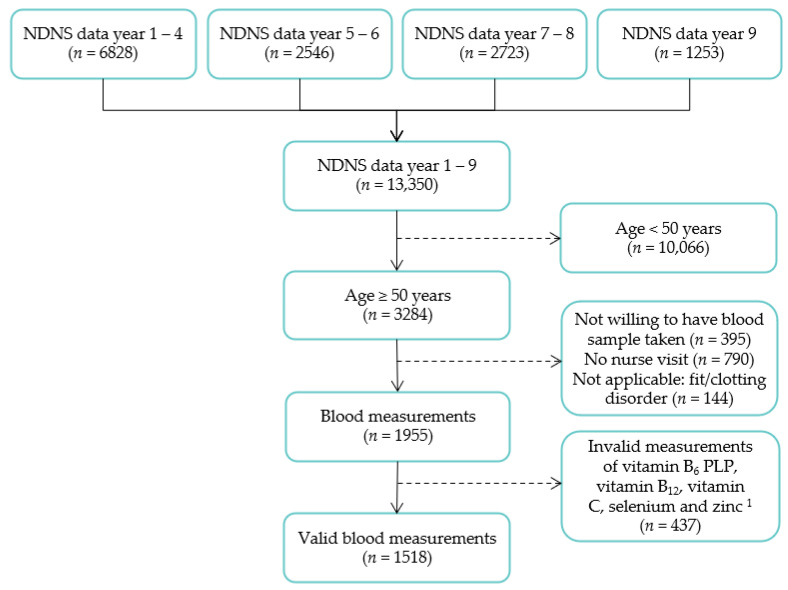
Flowchart of the study population for analyses. ^1^ Reasons for invalid measurements included an incomplete blood volume, notional results, incorrect labelling and insufficient sample for analysis.

**Figure 3 nutrients-13-01883-f003:**
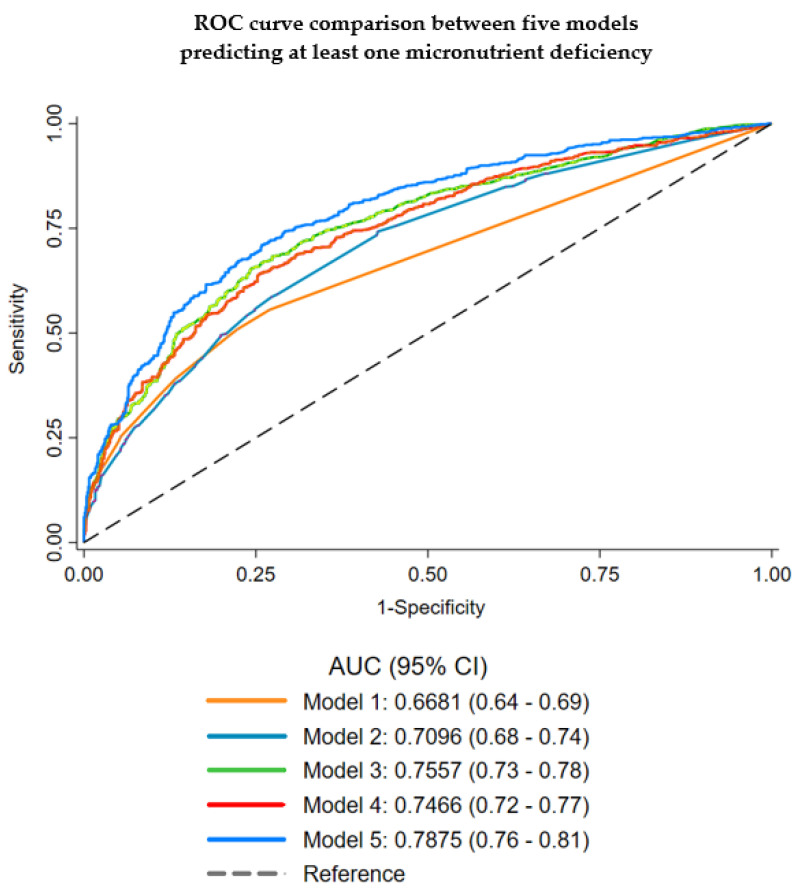
Area under the ROC curves (AUC) for five different predictive models of micronutrient deficiency showing discriminative ability (chi-square test *p* ≤ 0.001). Model 1: Routine biochemical diagnostic tests (total cholesterol, haemoglobin, ferritin, HbA1c, CRP and 25-hydroxy vitamin D). Model 2: Established malnutrition indicators (individual components of established malnutrition screening tools) (number of medicines, any dietary-supplement use in the last year, appetite, self-assessed general health, and fruit and vegetable intake). Model 3: Routine biochemical diagnostic tests + covariates (total cholesterol, haemoglobin, ferritin, HbA1c, CRP, 25-hydroxy vitamin D, sex, age category, ethnic group, region, qualification and smoking status). Model 4: Established malnutrition indicators + covariates (number of medicines, any dietary-supplement use in the last year, appetite, self-assessed general health, fruit and vegetable intake, sex, age category, ethnic group, region, qualification and smoking status). Model 5: Routine biochemical diagnostic tests + established malnutrition indicators + covariates (final prediction model) (total cholesterol, haemoglobin, ferritin, HbA1c, CRP, 25-hydroxy vitamin D, number of medicines, any dietary-supplement use in the last year, appetite, self-assessed general health, fruit and vegetable intake, sex, age category, ethnic group, region, qualification and smoking status).

**Table 1 nutrients-13-01883-t001:** Characteristics of study population according to micronutrient deficiency biomarker status (vitamin B_6_ PLP, vitamin B_12_, vitamin C and selenium, zinc) (*n* = 1518).

Characteristic	No Micronutrient Deficiencies (*n* = 789) *	At Least One Micronutrient Deficiency (*n* = 729) *	*p*-Value ^1^
Sex, *women*	462 (58.6)	406 (55.7)	0.260
Age group			<0.001
50–59 years	353 (44.7)	223 (30.6)	
60–69 years	267 (33.8)	255 (35.0)	
≥70 years	169 (21.4)	251 (34.4)	
Ethnic group, *white British*	759 (96.2)	708 (97.1)	0.319
Region			<0.001
England—North	153 (19.4)	140 (19.2)	
England—Central/Midlands	102 (12.9)	75 (10.3)	
England—South	271 (34.4)	184 (25.2)	
Scotland	113 (14.3)	128 (17.6)	
Wales	96 (12.2)	151 (20.7)	
Northern Ireland	54 (6.8)	51 (7.0)	
Qualification			<0.001
Secondary education or less	336 (42.6)	415 (56.9)	
Further education	105 (13.3)	91 (12.5)	
Higher education	309 (39.2)	172 (23.6)	
Other	39 (4.9)	51 (7.0)	
Marital status			<0.001
Single, never married	81 (10.3)	57 (7.8)	
Married or partnership	495 (62.7)	392 (53.8)	
Divorced or widowed	213 (27.0)	280 (38.4)	
Smoking status (cigarettes)			<0.001
Never smoker	483 (61.2)	349 (47.9)	
Former smoker	251 (31.8)	234 (32.1)	
Current smoker	55 (7.0)	146 (20.0)	
Self-assessed general health			<0.001
Good	654 (82.9)	466 (63.9)	
Fair	122 (15.5)	204 (28.0)	
Bad	13 (1.7)	59 (8.1)	
Has longstanding illness, *yes*	354 (44.9)	441 (60.5)	<0.001
Number of medicines			<0.001
No medication	290 (36.8)	161 (22.1)	
1–4 medicines	389 (49.3)	319 (43.8)	
5 or more medicines	110 (13.9)	249 (34.2)	
Any dietary supplement use last year, *yes*	386 (48.9)	229 (31.4)	<0.001
Any of own teeth, *yes*	722 (91.5)	590 (80.9)	<0.001
Appetite			<0.001
Good	342 (43.4)	233 (32.0)	
Average	132 (16.7)	134 (18.4)	
Poor	7 (0.9)	42 (5.8)	
N/A to survey year	308 (39.0)	729 (43.9)	
BMI (kg/m^2^), mean ± SD ^2^	27.7 ± 4.6	28.9 ± 5.5	<0.001
BMI (kg/m^2^)			<0.001
≥20 (age < 70 years) or ≥22 (age ≥ 70 years)	739 (93.7)	636 (87.2)	
<20 (age < 70 years) or <22 (age ≥ 70 years)	23 (2.9)	30 (4.1)	
Unknown	27 (3.4)	63 (8.6)	
Protein intake (g)			<0.001
≥RNI	653 (82.8)	451 (61.9)	
<RNI	115 (14.6)	227 (31.1)	
Unknown	21 (2.7)	51 (7.0)	
Energy intake (kcal)			0.198
≥EAR	128 (16.2)	101 (13.9)	
<EAR	661 (83.8)	628 (86.2)	
Protein intake (g) and energy intake (kcal)			<0.001
≥RNI and ≥EAR	126 (16.0)	91 (12.5)	
<RNI and <EAR	115 (14.6)	223 (30.6)	
Either <RNI or <EAR	527 (66.8)	364 (49.9)	
Unknown	21 (2.7)	51 (7.0)	
Fruit and vegetable intake ^3^			
<5 portions (80 g)/day	430 (54.5)	514 (70.5)	<0.001
<2 portions (80 g)/day	66 (8.4)	156 (21.4)	<0.001
Fluid intake ^3^			
<1600 mL/day (women) and <2000 mL/day (men)	438 (55.5)	470 (64.5)	<0.001
<1250 mL/day	174 (22.1)	187 (25.7)	0.100
<750 mL/day	21 (2.7)	23 (3.2)	0.567

* Data are presented as mean ± SD or number (%). ^1^ Chi-square test for categorical variables and ANOVA test for continuous variables. ^2^ Based on 762 valid observations for no micronutrient deficiencies and 666 valid observations for ≥1 micronutrient deficiency. ^3^ No unknown categories, remaining number (%) of subjects are ≥ the cut-off point. BMI, body mass index; RNI, Recommended Nutrient Intake; EAR, Estimated Average Requirement.

**Table 2 nutrients-13-01883-t002:** Descriptives of the micronutrient deficiency biomarkers selected to define a poor nutritional status ^1^.

Micronutrient Deficiency Biomarkers	Cut-Off Point Inadequate Status	Sex	*n* Total	Mean ± SD	*n* (%) Inadequate Status
Vitamin B6 PLP (nmol/L) ^2^	<30 [58]	Total	1518	52.0 ± 42.7	454 (29.9)
		Men	650	48.9 ± 32.5	191 (29.4)
		Women	868	54.3 ± 48.8	263 (30.3)
Selenium (µmol/L)	<0.9 [59]	Total	1518	1.06 ± 0.24	338 (22.3)
		Men	650	1.04 ± 0.22	151 (23.2)
		Women	868	1.07 ± 0.25	187 (21.5)
Zinc (µmol/L)	<11 [44]	Total	1518	13.46 ± 2.55	177 (11.7)
		Men	650	13.60 ± 2.57	77(11.8)
		Women	868	13.36 ± 2.53	100 (11.5)
Vitamin B12 (pmol/L)	<150 [60]	Total	1518	271.5 ± 103.0	76 (5.0)
		Men	650	254.7 ± 87.1	39 (6.0)
		Women	868	284.1 ± 111.8	37 (4.3)
Vitamin C (µmol/L)	<11.4 [61,62]	Total	1518	50.1 ± 21.8	56 (3.7)
		Men	650	45.2 ± 20.1	28 (4.3)
		Women	868	53.8 ± 22.4	28 (3.2)

Data are presented as mean ± SD or number (%). ^1^ Selection process described in methods. ^2^ Data are not normally distributed.

**Table 3 nutrients-13-01883-t003:** Descriptives of routine biochemical diagnostic tests for subjects with a valid measurement and selected cut-off points to test relationships with poor nutritional status markers.

Routine Biochemical Diagnostic Test	At Risk Cut-Off Point	Sex	*n* Valid Result ^2^	Mean ± SD	*n* (%) at Risk	*n* Valid Result ^3^	Mean ± SD No Deficiencies (*n* = 789)	*n* Valid Result ^3^	Mean ± SD ≥ 1 Deficiency (*n* = 729)	*p*-Value ^4^
Total Cholesterol (mmol/L)	<4.1 [45,46]	Total	1490	5.28 ± 1.17	223 (15.0)	776	5.46 ± 1.09	714	5.08 ± 1.22	<0.001 *
		Men	638	4.89 ± 1.11	152 (23.8)					
		Women	852	5.57 ± 1.13	71 (8.3)					
Triglycerides (mmol/L) ^1^	<0.5 [44]	Total	1483	1.37 ± 0.81	14 (0.9)	774	1.30 ± 0.70	709	1.44 ± 0.92	0.001 *
		Men	637	1.45 ± 0.93	4 (0.6)					
		Women	846	1.31 ± 0.71	10 (1.2)					
LDL (mmol/L)	<2.2 [32]	Total	1472	3.20 ± 1.04	255 (17.3)	769	3.34 ± 0.97	703	3.05 ± 1.08	<0.001 *
		Men	631	2.98 ± 1.00	151 (23.9)					
		Women	841	3.37 ± 1.03	104 (12.4)					
HDL (mmol/L)	<1.0 [44] <1.2 [44]	Total	1490	1.49 ± 0.47	277 (18.6)	324 452	1.35 ± 0.38 1.71 ± 0.47	314 400	1.25 ± 0.38 1.55 ± 0.46	0.001 * <0.001 *
		Men	638	1.30 ± 0.38	131 (20.5)					
		Women	852	1.63 ± 0.47	146 (17.1)					
Haemoglobin (g/dL)	<13 [44] <12 [44]	Total	1436	13.8 ± 1.3	131 (9.1)	313 438	14.8 ± 1.1 13.4 ± 0.9	304 381	14.3 ± 1.4 13.1 ± 1.2	<0.001 * 0.006 *
		Men	617	14.6 ± 1.3	55 (8.9)					
		Women	819	13.3 ± 1.1	76 (9.3)					
Haematocrit (%)	<40 [44] <36 [44]	Total	1436	42.0 ± 4.2	164 (11.4)	313 438	44.8 ± 3.5 40.5 ± 3.0	304 381	43.3 ± 4.4 40.2 ± 4.0	<0.001 * 0.256
		Men	617	44.1 ± 4.0	85 (13.8)					
		Women	819	40.4 ± 3.5	79 (9.6)					
Mean Cell Volume (fL)	<83 or >101 [44]	Total	1436	93.8 ± 5.6	147 (10.2)	751	93.8 ± 5.0	685	93.9 ± 6.1	0.620
		Men	617	94.3 ± 5.8	77 (12.5)					
		Women	819	93.5 ± 5.4	70 (8.5)					
Ferritin (µg/L) ^1^	<23 [44]	Total	1512	118.1 ± 126.4	139 (9.2)	787	120.9 ± 115.2	725	115.1 ± 137.6	0.372
		Men	649	153.2 ± 152.5	36 (5.5)					
		Women	863	91.7 ± 94.4	103 (11.9)					
HbA1c (%) ^1^	<5.0 [28,49]	Total	1429	5.8 ± 0.8	34 (2.4)	739	5.7 ± 0.7	690	5.9 ± 0.9	<0.001 *
		Men	609	5.9 ± 0.9	13 (2.1)					
		Women	820	5.8 ± 0.6	21 (2.6)					
Lymphocyte Count (10^9^/L)	<1.0 [44]	Total	1337	1.92 ± 0.69	69 (5.2)	707	1.94 ± 0.69	630	1.91 ± 0.69	0.499
		Men	568	1.85 ± 0.63	34 (6.0)					
		Women	769	1.98 ± 0.73	35 (4.6)					
White Blood Cell Count (10^9^/L)	<4.0 [44]	Total	1435	6.32 ± 2.28	59 (4.1)	751	6.04 ± 2.53	684	6.63 ± 1.93	<0.001 *
		Men	616	6.41 ± 1.67	16 (2.6)					
		Women	819	6.26 ± 2.65	43 (5.3)					
CRP (mg/L) ^1^	>10 [44]	Total	1300	4.44 ± 6.63	100 (7.7)	654	3.39 ± 5.29	646	5.50 ± 7.61	<0.001 *
		Men	552	3.94 ± 5.63	32 (5.8)					
		Women	748	4.80 ± 7.26	68 (9.1)					
eGFR (mL/min/1.73 m^2^)	<60 [47]	Total	1505	76.6 ± 16.9	241 (16.0)	785	78.5 ± 15.0	720	74.4 ± 18.6	<0.001 *
		Men	646	76.7 ± 16.3	91 (14.1)					
		Women	859	76.5 ± 17.3	150 (17.5)					
Creatinine (µmol/L)	<59 [44] <45 [44]	Total	1505	82.9 ± 24.7	14 (0.9)	326 459	91.5 ± 14.6 73.8 ± 27.1	320 400	95.7 ± 25.1 76.2 ± 21.3	0.010 * 0.148
		Men	646	93.6 ± 20.6	6 (0.9)					
		Women	859	74.9 ± 24.6	8 (0.9)					
25-Hydroxy Vitamin D (nmol/L)	<25 [48]	Total	1481	47.8 ± 20.4	200 (13.5)	773	52.1 ± 19.9	708	43.1 ± 19.9	<0.001 *
		Men	632	48.2 ± 19.9	77 (12.2)					
		Women	849	47.5 ± 20.8	123 (14.5)					

Data are presented as mean ± SD or number (%). ^1^ Data are not normally distributed. ^2^ Number of observations available with a valid measurement for the specified biochemical, separated by sex. ^3^ Number of observations available with a valid measurement for the specified biochemical, separated by micronutrient deficiency group. ^4^ ANOVA test; frequently unequal variances between groups. * Statistically significant difference between means. LDL, low-density lipoproteins; HDL, high-density lipoproteins; HbA1c, haemoglobin A1c; CRP, C-reactive protein; eGFR, Estimated Glomerular Filtration Rate.

**Table 4 nutrients-13-01883-t004:** Multiple univariate and multivariable logistic regression analyses of poor nutritional status ^1^ as dependent binary variable and each routine biochemical diagnostic test individually as independent binary variable, in unadjusted and adjusted models ^2^ (Stages 1a and 1b).

Low Concentrations of Routine Biochemical Diagnostic Tests	Univariate Analysis ≥ 1 Micronutrient Deficiency vs. No Micronutrient Deficiencies				Multivariable Analysis ≥ 1 Micronutrient Deficiency vs. No Micronutrient Deficiencies	
	Crude OR (95% CI)	*p*-Value	Adjusted ^3^ OR (95% CI)	*p*-Value	Adjusted ^3^ OR (95% CI)	*p*-Value
Total Cholesterol < 4.1 mmol/L	2.48 (1.84–3.35)	<0.001 *	2.29 (1.66–3.15)	<0.001 *	2.03 (1.44–2.88)	<0.001 *
Triglycerides < 0.5 mmol/L	0.81 (0.28–2.35)	0.698	1.05 (0.34–3.20)	0.938	1.14 (0.34–3.86)	0.832
LDL < 2.2 mmol/L	2.13 (1.62–2.82)	<0.001 *	1.84 (1.37–2.48)	<0.001 *	- ^4^	-
HDL < 1.0 mmol/L (men) and < 1.2 mmol/L (women)	1.71 (1.31–2.22)	<0.001 *	1.56 (1.17–2.07)	0.002 *	1.17 (0.85–1.60)	0.332
Haemoglobin < 13 g/dL (men) and < 12 g/dL (women)	4.26 (2.78–6.53)	<0.001 *	4.24 (2.72–6.61)	<0.001 *	2.71 (1.68–4.37)	<0.001 *
Haematocrit < 40% (men) and < 36% (women)	2.74 (1.93–3.88)	<0.001 *	2.65 (1.83–3.82)	<0.001 *	- ^4^	-
Mean Cell Volume < 83 fL or > 10 ^1^ fL	1.60 (1.13–2.25)	0.008 *	1.43 (0.99–2.06)	0.058	1.17 (0.78–1.75)	0.456
Ferritin < 23 µg/L	2.06 (1.43–2.95)	<0.001 *	2.59 (1.76–3.82)	<0.001 *	2.25 (1.49–3.41)	<0.001 *
HbA1c < 5.0%	1.77 (0.88–3.56)	0.109	2.49 (1.20–5.19)	0.015 *	2.97 (1.38–6.40)	0.006 *
Lymphocyte Count < 1.0 × 10^9^/L	1.34 (0.83–2.18)	0.232	1.30 (0.78–2.17)	0.322	0.86 (0.48–1.55)	0.611
White Blood Cell Count < 4.0 × 10^9^/L	0.91 (0.54–1.53)	0.723	1.14 (0.65–1.98)	0.650	1.11 (0.60–2.05)	0.738
CRP > 10 mg/L	5.07 (3.04–8.44)	<0.001 *	5.02 (2.96–8.53)	<0.001 *	5.18 (3.00–8.95)	<0.001 *
eGFR < 60 mL/min/1.73 m^2^	2.30 (1.73–3.07)	<0.001 *	1.67 (1.21–2.31)	0.002 *	1.45 (1.02–2.05)	0.037 *
Creatinine < 59 µmol/L (men) and <45 µmol/L (women)	14.31 (1.87–110)	0.010 *	11.85 (1.50–93.9)	0.019 *	- ^4^	-
25-Hydroxy Vitamin D < 25 nmol/L	3.31 (2.38–4.60)	<0.001 *	2.94 (2.07–4.16)	<0.001 *	2.93 (2.04–4.22)	<0.001 *

^1^ Poor nutritional status is defined as the presence of at least one micronutrient deficiency. ^2^ See Table 3 for number of subjects available for each biochemical analysis. ^3^ Adjusted for sex, age category, ethnic group, region, qualification and smoking status. ^4^ Excluded from multivariable analysis due to high correlation with another routine biochemical diagnostic test. * Statistically significant. OR, odds ratio; CI, confidence interval; LDL, low-density lipoproteins; HDL, high-density lipoproteins; HbA1c, haemoglobin A1c; CRP, C-reactive protein; eGFR, Estimated Glomerular Filtration Rate.

**Table 5 nutrients-13-01883-t005:** Multiple univariate logistic regression analyses of the established malnutrition indicators as dependent binary variables and each routine biochemical diagnostic test individually as independent binary variable, unadjusted and adjusted ^1^ (Stage 2).

Low Concentrations of Routine Biochemical Diagnostic Tests	Protein Intake (g) < RNI (UK DRV [54])		Energy Intake (kcal) < EAR (SACN [55])		Fruit and Vegetable Intake < 2 Portions/Day (MNA)		Fluid Intake < 2000 mL/Day (men) and <1600 mL/Day (Women) (EFSA [57])		BMI < 20 kg/m^2^ (age < 70 years) and <22 kg/m^2^ (Age ≥ 70 years) (GLIM [53])	
Crude OR (95% CI) and *p*-value										
Total Cholesterol < 4.1 mmol/L	1.59 (1.15–2.19)	0.005 *	1.12 (0.74–1.68)	0.593	1.39 (0.95–2.01)	0.086	1.68 (1.24–2.28)	0.001 *	1.25 (0.60–2.59)	0.558
Haemoglobin < 13 g/dL (men) and < 12 g/dL (women)	1.74 (1.17–2.60)	0.007 *	1.30 (0.76–2.24)	0.338	1.50 (0.95–2.37)	0.079	2.03 (1.35–3.04)	0.001 *	2.42 (1.15–5.10)	0.020 *
Ferritin < 23 µg/L	1.40 (0.94–2.08)	0.096	1.14 (0.69–1.89)	0.611	1.39 (0.88–2.18)	0.155	0.94 (0.66–1.34)	0.727	2.15 (1.02–4.51)	0.043 *
HbA1c < 5.0%	0.90 (0.39–2.10)	0.811	1.86 (0.56–6.12)	0.310	0.17 (0.02–1.27)	0.085	0.52 (0.26–1.04)	0.063	- ^3^	-
CRP > 10 mg/L	2.41 (1.55–3.73)	<0.001 *	0.80 (0.47–1.35)	0.400	1.50 (0.90–2.51)	0.118	0.85 (0.56–1.27)	0.421	1.25 (0.44–3.56)	0.670
eGFR < 60 mL/min/1.73 m^2^	1.70 (1.25–2.32)	0.001 *	1.21 (0.81–1.80)	0.361	1.33 (0.92–1.92)	0.124	2.02 (1.49–2.74)	<0.001 *	1.46 (0.74–2.88)	0.274
25-Hydroxy Vitamin D < 25 nmol/L	2.56 (1.85–3.53)	<0.001 *	1.70 (1.04–2.75)	0.033 *	2.53 (1.78–3.60)	<0.001 *	1.54 (1.12–2.12)	0.007 *	1.38 (0.66–2.88)	0.385
Adjusted ^2^ OR (95% CI) and *p*-value										
Total Cholesterol < 4.1 mmol/L	1.90 (1.35–2.69)	<0.001 *	1.52 (0.99–2.34)	0.057	1.31 (0.87–1.98)	0.197	1.23 (0.88–1.71)	0.222	1.04 (0.47–2.30)	0.924
Haemoglobin < 13 g/dL (men) and < 12 g/dL (women)	1.83 (1.20–2.80)	0.005 *	1.41 (0.80–2.48)	0.232	1.6 (1.02–2.75)	0.041 *	1.58 (1.03–2.43)	0.035 *	1.73 (0.78–3.87)	0.178
Ferritin < 23 µg/L	1.36 (0.91–2.04)	0.136	0.99 (0.59–1.67)	0.984	1.66 (1.02–2.71)	0.040 *	1.06 (0.73–1.54)	0.753	2.57 (1.17–5.67)	0.019 *
HbA1c < 5.0%	0.94 (0.40–2.22)	0.885	1.74 (0.52–5.81)	0.370	0.17 (0.02–1.30)	0.088	0.59 (0.29–1.21)	0.152	- ^3^	-
CRP > 10 mg/L	2.22 (1.41–3.50)	0.001 *	0.71 (0.413–1.23)	0.226	1.21 (0.70–2.10)	0.502	0.77 (0.50–1.19)	0.241	0.88 (0.30–2.62)	0.820
eGFR < 60 mL/min/1.73 m^2^	1.74 (1.22–2.48)	0.002 *	1.29 (0.83–2.01)	0.264	1.01 (0.67–1.55)	0.946	1.50 (1.07–2.12)	0.019 *	0.50 (0.24–1.06)	0.069
25-Hydroxy Vitamin D < 25 nmol/L	2.35 (1.68–3.27)	<0.001 *	1.61 (0.98–2.65)	0.061	2.03 (1.38–2.97)	<0.001 *	1.45 (1.03–2.03)	0.031 *	0.99 (0.46–2.17)	0.989

^1^ See Table 3 for number of subjects available for each biochemical analysis, and Table 6 for number of subjects available for each established malnutrition indicator. ^2^ Adjusted for sex, age category, ethnic group, region, qualification and smoking status. ^3^ Analysis error: no subjects with a low BMI and low HbA1c present in the data. * Statistically significant. OR, odds ratio; CI, confidence interval; HbA1c, haemoglobin A1c; CRP, C-reactive protein; eGFR, Estimated Glomerular Filtration Rate; RNI, Reference Nutrient Intake; DRV, Dietary Reference Value; EAR, Estimated Average Requirement; SACN, Scientific Advisory Committee on Nutrition; MNA, Mini Nutritional Assessment; EFSA, European Food Safety Authority; GLIM, Global Leadership Initiative on Malnutrition.

**Table 6 nutrients-13-01883-t006:** Univariate logistic regression analyses of poor nutritional status ^1^ as dependent binary variable and established malnutrition indicators as independent binary variables, adjusted for covariates (Stage 3).

Low Levels of Established Malnutrition Indicators (Source Cut-Off Point)	≥1 Micronutrient Deficiency vs. No Micronutrient Deficiencies		
	*n* Analysis	Adjusted ^2^ OR (95% CI)	*p*-Value
Protein intake (g) < RNI (UK DRV [54])	1446	2.82 (2.25–3.70)	<0.001 *
Energy intake (kcal) < EAR (SACN [55])	1518	1.21 (0.89–1.64)	0.216
Energy intake (kcal) < 1800 kcal/day	1518	1.56 (1.22–1.99)	<0.001 *
Protein intake (g) < RNI and energy intake (kcal) < EAR	555	2.53 (1.71–3.73)	<0.001 *
Fruit and vegetable intake < 5 portions/day (Eatwell Guide [56])	1518	1.66 (1.32–2.08)	<0.001 *
Fruit and vegetable intake < 2 portions/day (MNA)	1518	2.12 (1.52–2.95)	<0.001 *
Fluid intake < 2000 mL/day (men) and <1600 mL/day (women) (EFSA [57])	1518	1.27 (1.01–1.59)	0.040 *
Fluid intake < 1250 mL/day (MNA)	1518	1.04 (0.80–1.34)	0.773
Fluid intake < 750 mL/day (MNA)	1518	0.98 (0.52–1.86)	0.962
BMI < 20 kg/m^2^ (age < 70 years) and <22 kg/m^2^ (age ≥ 70 years) (GLIM [53])	1428	0.93 (0.51–1.68)	0.803

^1^ Poor nutritional status is defined as the presence of at least one micronutrient deficiency. ^2^ Adjusted for sex, age category, ethnic group, region, qualification and smoking status. * Statistically significant. OR, odds ratio; CI, confidence interval; RNI, Reference Nutrient Intake; DRV, Dietary Reference Value; EAR, Estimated Average Requirement; SACN, Scientific Advisory Committee on Nutrition; MNA, Mini Nutritional Assessment; EFSA, European Food Safety Authority; GLIM, Global Leadership Initiative on Malnutrition.

**Table 7 nutrients-13-01883-t007:** Final model predicting a poor nutritional status (presence of at least one micronutrient deficiency) (*n* = 1518) ^1^, resulting from stepwise backward selection in a logistic regression analysis including all routine biochemical diagnostic tests, established malnutrition indicators and covariates. Covariates were locked in the model (Stage 4).

Predictors	At Least One Micronutrient Deficiency vs. No Micronutrient Deficiencies	
	Adjusted ^2^ OR (95% CI)	*p*-Value
Routine biochemical diagnostic tests (proposed tools for identifying a poor nutritional status)		
Total Cholesterol < 4.1 mmol/L	1.70 (1.19–2.43)	0.003 *
Haemoglobin < 13 g/dL (men) and <12 g/dL (women)	2.45 (1.50–4.01)	<0.001 *
Ferritin < 23 µg/L	2.28 (1.49–3.49)	<0.001 *
HbA1c < 5.0%	2.99 (1.39–6.41)	0.005 *
CRP > 10 mg/L	4.71 (2.70–8.22)	<0.001 *
25-Hydroxy Vitamin D < 25 nmol/L	2.43 (1.67–3.54)	<0.001 *
Established malnutrition indicators (individual components of established malnutrition screening tools/risk factors)		
Number of medicines		
1–4 medicines vs. no medication	1.26 (0.95–1.67)	0.109
5 or more medicines vs. no medication	2.07 (1.40–3.06)	<0.001 *
Any dietary supplement use last year, yes vs. no	0.50 (0.39–0.64)	<0.001 *
Appetite ^3^		
Average vs. good	0.94 (0.68–1.29)	0.705
Poor vs. good	2.85 (1.17–6.98)	0.022 *
Self-assessed general health		
Fair vs. good	1.20 (0.88–1.65)	0.251
Bad vs. good	2.44 (1.20–4.96)	0.014 *
Fruit and vegetable, <2 portions/day vs. 2 or more portions/day	1.62 (1.13–2.33)	0.009 *
Covariates (locked into model)		
Sex, women vs. men	0.86 (0.67–1.10)	0.230
Age group		
60–69 years vs. 50–59 years	1.40 (1.05–1.86)	0.020 *
≥70 years vs. 50–59 years	2.07 (1.50–2.85)	<0.001 *
Ethnic group, White British vs. non-white	1.05 (0.54–2.04)	0.889
Region		
England—North vs. England—Central/Midlands	1.20 (0.78–1.85)	0.407
England—South vs. England—Central/Midlands	1.07 (0.71–1.61)	0.754
Scotland vs. England—Central/Midlands	1.31 (0.83–2.06)	0.246
Wales vs. England—Central/Midlands	2.30 (1.46–3.61)	<0.001 *
Northern Ireland vs. England—Central/Midlands	1.02 (0.58–1.80)	0.949
Qualification		
Further education vs. secondary education or less	1.04 (0.72–1.50)	0.834
Higher education vs. secondary education or less	0.78 (0.59–1.03)	0.085
Other vs. secondary education or less	1.15 (0.68–1.93)	0.601
Smoking status (cigarettes)		
Former smoker vs. never smoker	1.10 (0.84–1.43)	0.491
Current smoker vs. never smoker	3.17 (2.14–4.69)	<0.001 *

^1^ Hosmer–Lemeshow χ^2^ = 7.22, *p* = 0.513, c-statistic (AUC) = 0.79 (95% CI: 0.76–0.81); McFadden’s pseudo r-squared, 0.20; Brier score, 0.19; sensitivity, 66.0%; specificity, 78.1%; positive predictive value, 73.6%; negative predictive value, 71.3%. ^2^ Adjusted for the covariates sex, age category, ethnic group, region, qualification and smoking status. ^3^ Substantial less subjects available for analysis (*n* = 890). * Statistically significant. HbA1c, haemoglobin A1c; CRP, C-reactive protein.

## Data Availability

The NDNS data used in our analyses are available to researchers on application to The UK Data Service, which is funded by the UKRI through the ESRC with contributions from the University of Essex, the University of Manchester and Jisc. Weblink: https://www.ukdataservice.ac.uk/(accessed on 31 March 2021).

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
