# Peer review of "Predicting Malnutrition Risk with Data from Routinely Measured Clinical Biochemical Diagnostic Tests in Free-Living Older Populations"

_nutrients, 2021, doi:10.3390/nu13061883_

Round 1

Reviewer 1 Report

The manuscript by Trujen et al., describes the use of readily available clinical blood measurements from routine biochemical diagnostic tests to predict nutritional status. They observe that cholestrol level, haemoglobin, ferritin and vitamin D or high C-reactive protein are good predictors of poor nutrition. This work is essentially important as it can facilitate early detection/prediction of malnutrition in the elderly population. The authors explain the data well and their findings are well summarized in table format and limitations of the study are addressed.

A few minor things that the authors should consider addressing are;

1). In the study design section, the authors indicate that they selected a sample size of ~1,000 people with 500 adults and 500 children each year. The authors do not indicate why they excluded children in the report. Would the same indicators be used to predict poor nutrition in children as well?

2) Under qualification, secondary education or less had more people with at least one micronutrient deficiency. Did the author consider if type of profession (skilled/non-skilled) contributed to this?

3) Legend for figure 3 some line break off to start a new line.

Reviewer 2 Report

Early detection is very important to preven tmalnutrition in the elderly. This study is a very important paper  that examines whether readily available and routinely measured data  can predict deterioration in nutritional status.
Methods and number of participants are reasonable. However, the discussion  of the clinical biochemical diagnostic test data used is somewhat inadequate.

Line 573-: 25-hydroxy vitamin D is also related to time spent in the sun. However, the authors did not mention this in their discussion. The authors should include sun exposure in their discussion.

Line575: Low concentrations of HbA1c(%)  are <5.0, but this index is also an indicator of diabetes, and if it is too high, it can be a risk for hypnutrition. This should also be discussed in the discussion.

Line612-: Inflammation increases CRP concerntration, and decreases albumin levels, and CRP concerntration are an important indicator to determine whether malnutrition is due to diet or inflammation. Please emphasize this point a little more in your discussion.
